# Hydrogen-Rich Water Can Restrict the Formation of Biogenic Amines in Red Beet Pickles

**Duried Alwazeer** [1,2,3,*] **, Menekşe Bulut** [2,3,4] **and Yasemin Çelebi** [5]

1 Department of Nutrition and Dietetics, Faculty of Health Sciences, Igdir University, Igdir 76000, Turkey
2 Research Center for Redox Applications in Foods (RCRAF), Igdir University, Igdir 76000, Turkey
3 Innovative Food Technologies Development, Application and Research Center, Igdir University, Igdir 76000, Turkey
4 Department of Food Engineering, Faculty of Engineering, Igdir University, Igdir 76000, Turkey
5 Department of Food Processing, Eşme Vocational School, Uşak University, Uşak 64600, Turkey
* Correspondence: alwazeerd@gmail.com

**Abstract:** Fermented foods are considered the main sources of biogenic amines (BAs) in the human diet while lactic acid bacteria (LAB) are the main producers of BAs. Normal water (NW) and hydrogen-rich water (HRW) were used for preparing red beet pickles, i.e., NWP and HRWP, respectively. The formation of BAs, i.e., aromatic amines (tyramine, 2-phenylethylamine), heterocyclic amines (histamine, tryptamine), and aliphatic di-amines (putrescine), was analyzed in both beet slices and brine of NWPs and HRWPs throughout the fermentation stages. Significant differences in redox value (Eh7) between NWP and HRWP brine samples were noticed during the first and last fermentation stages with lower values found for HRWPs. Total mesophilic aerobic bacteria (TMAB), yeast–mold, and LAB counts were higher for HRWPs than NWPs for all fermentation stages. Throughout fermentation stages, the levels of all BAs were lower in HRWPs than those of NWPs, and their levels in brines were higher than those of beets. At the end of fermentation, the levels (mg/kg) of BAs in NWPs and HRWPs were, respectively: tyramine, 72.76 and 61.74 (beet) and 113.49 and 92.67 (brine), 2-phenylethylamine, 48.00 and 40.00 (beet) and 58.01 and 50.19 (brine), histamine, 67.89 and 49.12 (beet) and 91.74 and 70.92 (brine), tryptamine, 93.14 and 77.23 (beet) and 119.00 and 93.11 (brine), putrescine, 81.11 and 63.56 (beet) and 106.75 and 85.93 (brine). Levels of BAs decreased by (%): 15.15 and 18.35 (tyramine), 16.67 and 13.44 (2-phenylethylamine), 27.65 and 22.7 (histamine), 17.09 and 21.76 (tryptamine), and 21.64 and 19.5 (putrescine) for beet and brine, respectively, when HRW was used in pickle preparation instead of NW. The results of this study suggest that the best method for limiting the formation of BAs in pickles is to use HRW in the fermentation phase then replace the fermentation medium with a new acidified and brined HRW followed by a pasteurization process.

**Keywords:** red beet (*Beta vulgaris*) pickle; hydrogen-rich water; molecular hydrogen; biogenic amines; fermentation

## 1. Introduction

Red beet is an important source of phytochemicals such as vitamins (C and B), minerals (K, Fe, P, and Mg), and pigments (betalains) [1]. The pickles of red beets are rich sources of these bioactive compounds [2]. The fermentation of vegetables in pickle preparation occurs due to the growth of natural microflora found in raw materials especially *Leuconostoc mesenteroides*, *Lactobacillus plantarum*, *Lactobacillus brevis*, and *Pediococcus* [3].

Biogenic amines (BAs) are low molecular weight organic nitrogenous compounds formed during decarboxylation of amino acids and transamination of aldehydes and ketones [4]. Most BAs are bioactive, such as tyramine, 2-phenylethylamine, and histamine, and exert effects on the central nervous system or at a vascular level with pathogenic effects with different degrees of intensity, ranging from headaches to death. Other amines, such as putrescine and cadaverine, alter the organoleptic properties of food [5]. Fermented foods

have been considered the main sources of BAs in the human diet and lactic acid bacteria (LAB) are the main producers of BAs [6]. Although plant-based foods may be observed as low-risk products regarding their content of BAs, their fermented products such as pickles, wine, and beer were considered as having a potential risk of BAs [3]. The legal limits of biogenic amines in foods set by government agencies are available only for histamine and for fish and fish products. The maximum allowable level of histamine ranges between 200 and 400 mg/kg in the USA, Europe, China, Korea, Australia, and New Zealand [7].

Molecular hydrogen ($H_2$) is the smallest element and a colorless and tasteless gas that is classified as a food additive with the code E949 [8]. $H_2$ possesses several benefits when it is considered for health and food applications, which can be attributed to the known selective antioxidant, antiradical, anti-inflammatory, antiapoptosis, antistress, and anti-cancer properties [9]. Hydrogen-rich water (HRW) is ordinary water with extra hydrogen molecules infused into it [10]. HRW is prepared by different methods such as bubbling hydrogen gas in water, the addition of magnesium into the water, or water electrolysis. HRW is more convenient and safer than hydrogen gas in terms of practical applications. Recently, the effect of HRW on extending the shelf life of harvested products has been reported by some researchers. It has been emphasized that HRW application increased the antioxidant capacity by preventing endogenous ethylene production and thus it can delay the ripening and aging of kiwi fruit [11]. In addition, recently, the potential use of $H_2$ has been studied in different food products to preserve the antioxidant properties of fruits [8,12–14]; to prevent oxidative deterioration and extend shelf life of milk [15] and butter [16]; and to deaccumulate heavy metals in butter [17]. Recently, our team revealed the ability of molecular hydrogen in its gas state and water-enriched state to restrict the formation of biogenic amines in LAB-cultured butter [18] and MAP-packaged fish [19]. The findings of recent studies, especially those performed in our laboratory, show the advantageous application of HRW in food industries. The preservation of oxidation-sensitive ingredients such as vitamins, phenolics, flavonoids, anthocyanins, pigments, and unsaturated fatty acids is one of many benefits of using HRW in food preparation. Additionally, the use of HRW in the washing phase of raw materials may help in decreasing their heavy metal content.

The present study aims to evaluate the effect of using hydrogen-rich water in the preparation of red beet pickles on the formation of biogenic amines.

## 2. Materials and Methods

### 2.1. Red Beet Pickle Preparation

The red beet (*Beta vulgaris*) used in the study was obtained from a local vendor in the city of Iğdır. After the washing phase, red beets were peeled, sliced, and placed in 660 mL sterilized glass jars. Five percent (*w/v*) NaCl and 10% (*v/v*) grape vinegar (containing 5% acetic acid) were used in the preparation of brines. HRW brine was prepared by bubbling hydrogen gas (99.9%, Elite Gaz Teknolojileri, Ankara, Turkey) into normal water-based brine at 1 L/min for 3 min using a hose equipped with a needle [18]. Both normal water- and HRW-based brines were poured into previously beet slice-filled jars. Jars were stored in an incubator (UTEST, Siemens Simatich Hmi) at $20 \pm 2\ ^\circ$C for 56 days.

### 2.2. pH and Oxidoreduction Potential (Eh) Measurement

pH and Eh analyses were performed by direct immersion of electrodes in brine using an SP10 R (Consort, Turnhout, Belgium) pH electrode and an SP60X (Consort, Turnhout, Belgium) Eh electrode, respectively, and a data acquisition multiparameter interface (Consort Multiparameter Analyser C3040, Turnhout, Belgium). The measured oxidoreduction potential (Em) was used in the following equation to determine the Eh value:

$$Eh = Em + Er \qquad (1)$$

where Eh is the electrode potential value referring to the normal hydrogen electrode and Er is the potential of the reference electrode (Ag/AgCl). The Eh7 values were then calculated as follows:

$$Eh7 = Eh - 59(7\text{-}pHm) \tag{2}$$

where pHm is the measured pH value of the sample.

### 2.3. Microbiological Counts

A decimal dilution method was used for enumerating the microbial counts in pickles. Ten grams of pickle samples including 5 g of each pickle and brine was transferred to a filtered Stomacher bag and 90 mL of sterile peptone buffer solution was added. The bags containing samples were then placed in a Stomacher blender (Bag Mixer, 400, Puycapel, France) for 2 min for homogenization purposes. One milliliter of different dilutions for each sample was placed in a Petri dish followed by pouring an appropriate agar medium as follows: sterile malt extract agar (Merck, Rahway, NJ, USA) for total yeasts and molds; sterile plate count agar (Merck, Rahway, NJ, USA) for total mesophile aerobic bacteria (TMAB); and sterile MRS agar (Merck, Rahway, NJ, USA) for lactic acid bacteria were used as microbiological growth media.

The inoculated Petri plates were incubated at $25 \pm 1\ °C$ for 4–5 days (yeasts and molds), $37\ °C$ for 24–48 h (mesophilic aerobic bacteria), and $37\ °C$ for 72 h (LAB). The results were reported as the decimal logarithm of colony-forming units per gram (log CFU/g).

### 2.4. Biogenic Amine Analysis

Biogenic amines and reagents (HPLC-grade with purity 98–99%) were purchased from Sigma-Aldrich (St. Louis, MO, USA). The content of BAs was analyzed by a chromatographic method according to Bulut et al. [18]. The HPLC system consisted of a quaternary pump (Ultimate, 3000 Pump, Thermo Fisher Scientific, Carlsbad, CA, USA), a Dionex Ultimate 3000 Diode Array Detector, an Ultimate 3000 Column Compartment, and a computer running a Chromeleon package program. The HPLC column was Spherisorb ODS2 ($200\ \mu m$ and $4.60\ mm \times 200\ mm$) (Phenomenex, Torrance, CA, USA).

### 2.5. Statistical Analysis

The data were evaluated with analysis of variance (ANOVA) using the statistical package in the SPSS (Armonk, NY, USA) 20 computer program. Differences between means were determined using Duncan's multiple comparison test, and significance was determined at the $p < 0.05$ level. Experiments were performed in duplicate and analyses in triplicate.

## 3. Results

### 3.1. pH and Oxidoreduction Potential (Eh)

#### 3.1.1. pH Measurement

Table 1 shows that pH values of brine in NWPs and HRWPs changed significantly during different fermentation stages ($p < 0.05$). The pH value at the initial time was low for both NWPs and HRWPs at 3.37 and 3.95, respectively, due to the presence of acetic acid found in vinegar that had been added at the time of brine preparation. The addition of vinegar to the brine is generally applied in pickle manufacturing for restricting the growth of spoilage microorganisms and promoting the growth of LAB. At the end of the fermentation stage (56th day), it was noted that there was no significant difference in pH values between NWPs and HRWPs ($p > 0.05$) (Table 1 and Figure 1).

After one week of fermentation, the pH value decreased to 3.07 and 3.08 for NWPs and HRWPs, respectively ($p < 0.05$). In the 2nd week of fermentation, the pH value increased again to 3.91 and 3.79 for NWPs and HRWPs, respectively ($p < 0.05$). LAB produce organic acids, especially lactic acid and acetic acid, during growth, leading to a decrease in the pH value. The increase in pH value after two weeks of fermentation indicates the growth of non-LAB yeasts that can metabolize organic acids, leading to an increase in the pH value of the medium.

**Table 1.** Changes in Eh7 (mV) and pH in beet (Beta vulgaris) pickles during fermentation stages (days).

|  |  | 0th Day | 7th Day | 14th Day | 28th Day | 56th Day |
|---|---|---|---|---|---|---|
| Eh7 (mV) | NWP | −18.48 ± 2.17 eA | −7.87 ± 2.20 dA | +74.10 ± 4.42 cA | +143.25 ± 5.95 aA | +132.02 ± 7.19 bA |
|  | HRWP | −198.95 ± 9.90 eB | −57.28 ± 4.57 bB | +70.84 ± 2.83 cA | +121.15 ± 3.43 aA | +82.43 ± 6.44 bB |
| pH | NWP | 3.37 ± 0.06 cB | 3.07 ± 0.03 dA | 3.91 ± 0.02 aA | 3.75 ± 0.02 bB | 3.78 ± 0.02 bA |
|  | HRWP | 3.95 ± 0.03 aA | 3.08 ± 0.02 dA | 3.76 ± 0.01 cB | 3.85 ± 0.02 bA | 3.77 ± 0.01 cA |

a, b, c Different lowercase letters (in the same row) indicate the presence of significant differences ($p < 0.05$) between samples in the different fermentation periods. A, B, C Different uppercase letters (in the same column) indicate the presence of significant differences ($p < 0.05$) between samples in the same fermentation period.

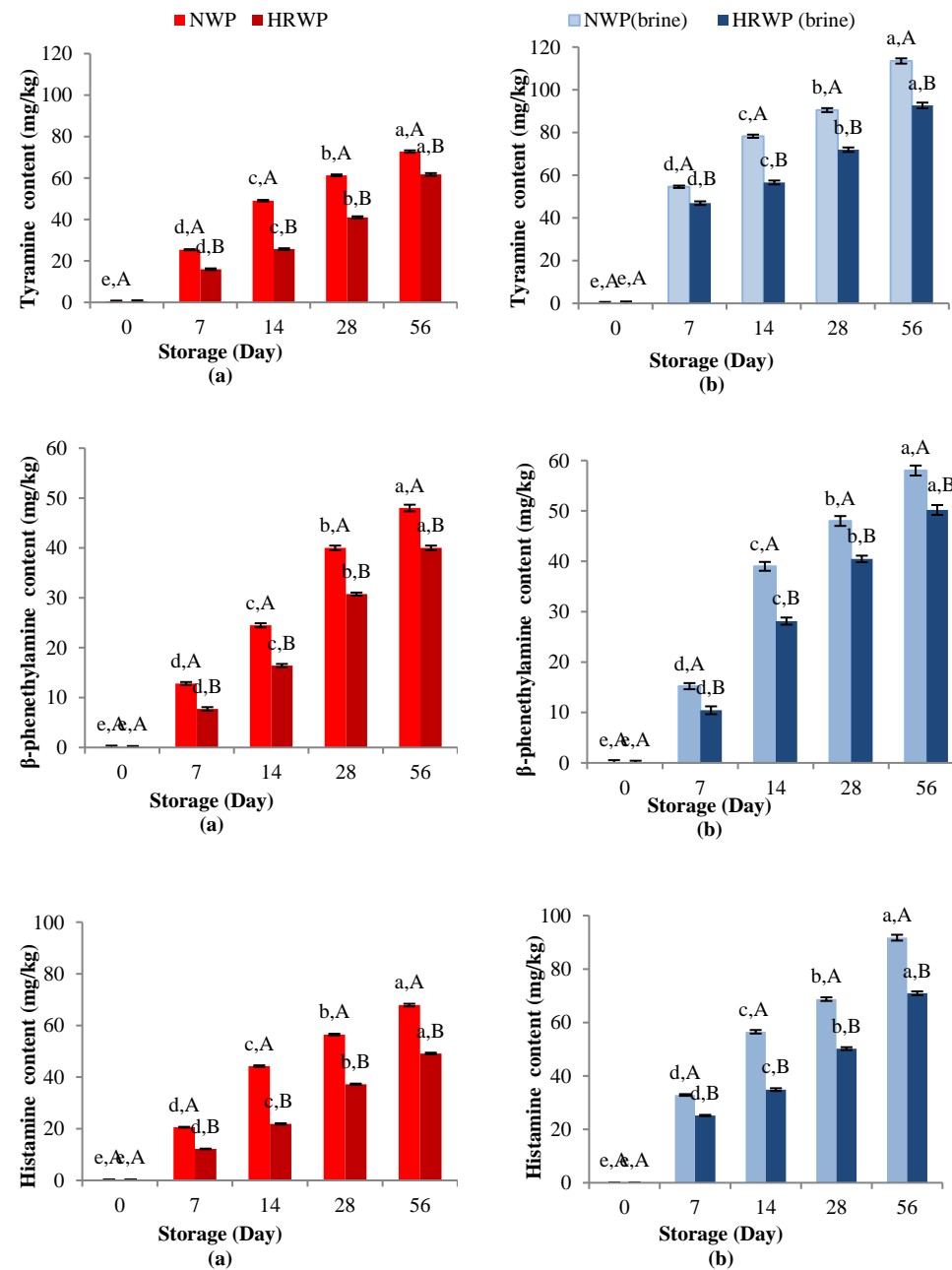

**Figure 1.** *Cont.*

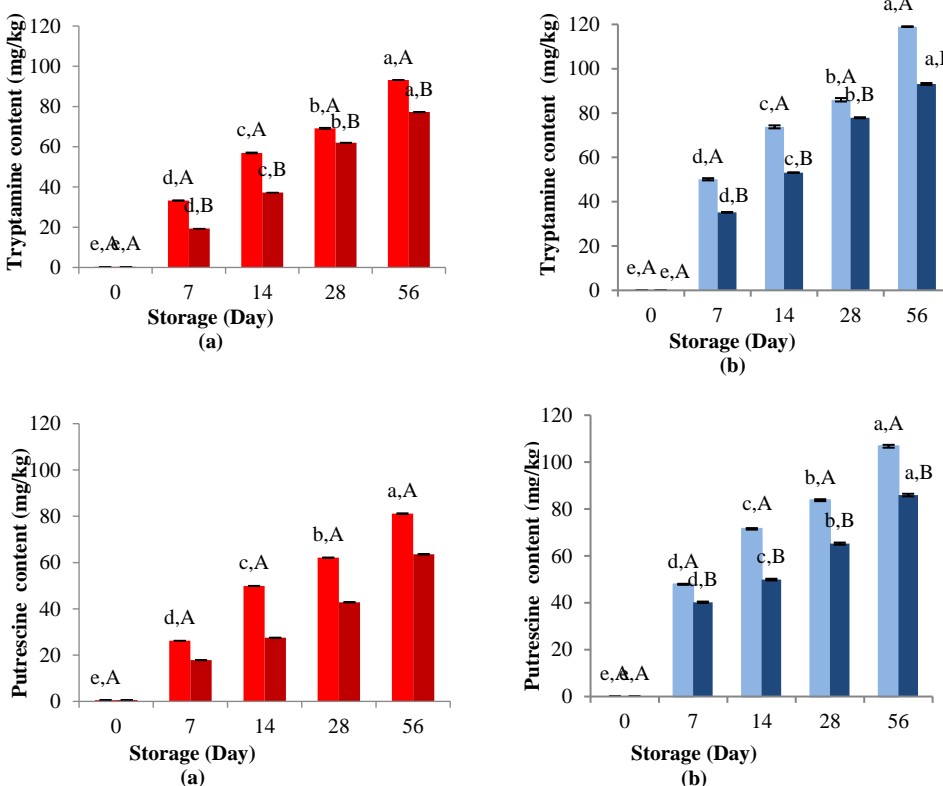

**Figure 1.** Biogenic amine content (mg/kg) in red beet (*Beta vulgaris*) (**a**) and brines (**b**).

Similar to our study, it was found that the pH value of cucumber pickles increased after 24 h of fermentation followed by a decreasing trend during the 2nd 24 h and then a stabilization phase up to the end of the fermentation period [20]. Generally, pH measurement is used during pickle production to monitor acid production and the growth of lactic acid bacteria [21].

### 3.1.2. Eh Measurement

The evolution of Eh7 value during fermentation stages (i.e., 0, 7, 14, 21, 28, and 56 days) is shown in Table 1. Eh7 value of brines of NWPs and HRWPs generally changed during fermentation stages ($p < 0.05$). While Eh7 values of NWP and HRWP brine samples were in negative ranges during the first 7 days of fermentation, they increased toward positive ranges after this point up to the end of the fermentation period. Similarly, Olsen (2008) found that Eh values of naturally fermented cucumber prepared by hydrogen-purged brine and control (non-purged) decreased during the first 24 h followed by an increase in the next 24 h and then a stabilization phase up to the end of the fermentation period [20]. Additionally, the Eh values of hydrogen-purged brine were reported to be lower than those of the control (non-purged).

Significant differences were shown between NWP and HRWP brine samples during the first and last fermentation stages ($p \leq 0.05$). Eh7 values of HRWP brine samples were significantly lower than those of NWP brine samples at all fermentation stages ($p < 0.05$). While the lowest and highest Eh7 values of NWP brine samples were −18 mV and +143 mV, those of HRWP brine samples were −198.95 mV and +121.15 mV, which corresponded to the initial and 4th week of fermentation, respectively (Table 1).

Eh is an important indicator of microbial growth and early prediction of spoilage yeast growth. Olsen and Pérez-Díaz (2009) reported that the Eh value of the fermentation medium of cucumber changed according to the microorganism used in the inoculation phase [21]. The authors measured the change in Eh value of fermentation medium of cucumber during two weeks where the medium was inoculated with pure and mixed cultures of a LAB

strain, i.e., *Lactobacillus plantarum*, a post-brining spoilage yeast, i.e., *Zygosaccharomyces globiformis*, as well as an early stage spoilage bacterium, i.e., *Enterobacter aerogenes*. They found that after 2 weeks of fermentation, the Eh value of jars inoculated with *L. plantarum*, *Z. globiformis*, and *E. aerogenes* was +453 ± 55, +104 ± 5, and −156 ± 73 mV, respectively, while cucumbers inoculated with a mixture of *L. plantarum* and *Z. globiformis* had an Eh value of +202 ± 24 mV. It is important to mention here that the authors used the measured value of Eh, not the corrected value, i.e., Eh7, which means that their Eh7 values should be lower by 88–206 mV than those cited in their paper.

The Eh value of a fermentation medium depends on different parameters such as the dissolved oxygen content, the chemical composition, especially the amounts of oxidants and reductants, the pH, and temperature [22]. The chemical composition of the medium depends mainly on the metabolites produced by the growth of microorganisms where every redox couple of any metabolite possesses a different reduction potential value. For example, the Eh of some biologically relevant redox couples such as NAD+/NADH, NADP+/NADPH, pyruvate, and H+/lactate is −316, −315, and −183 mV; while that of $O_2/H_2O$, ONOO-/$NO_2$•, and HO•, H+/$H_2O$ is +295, +1400, and +2310 mV, respectively [23]. The increase in the production of metabolites with reducing properties such as NADH and NADPH leads to a decrease in the Eh value, while the production of metabolites with oxidizing properties such as free radicals (ONOO-, HO•) leads to the increase in the Eh value of the medium. In an uncontrolled fermentation medium such as for NWPs, the Eh value initially decreases due to the growth of LAB and consumption of dissolved oxygen, then it increases due to cell death and liberation of free radicals (Olsen, 2008). Since the metabolite type changes with the microbial strain, the measurement of the Eh value of the fermentation medium can allow us to predict the type of microorganisms grown in the medium [21]. For example, due to the difference between the metabolites produced by spoilage yeasts and LAB during fermentation, the Eh value of the medium will change according to the dominant microorganism, which will allow for predicting the yeast spoilage [20]. Table 1 shows a significant difference in Eh7 values between NWP and HRWP brine samples in the first week of the fermentation period, and a difference in the growth of different microbial strains could be predicted. This means that the microflora of beet had developed differently in the HRWPs compared with NWPs.

### 3.2. Microbiological Counts

Table 2 shows microbiological counts of beet pickle samples. In all samples, TMAB, yeast–mold, and LAB counts changed significantly during the fermentation stages ($p < 0.05$) (Table 2).

**Table 2.** Microbial counts (log CFU/g) of pickles during fermentation stages (days).

|  |  | 0th Day | 7th Day | 14th Day | 28th Day | 56th Day |
|---|---|---|---|---|---|---|
| TMAB | NWP | 4.86 ± 0.03 eA | 8.28 ± 0.04 cB | 10.05 ± 0.07 aB | 8.94 ± 0.03 bB | 8.11 ± 0.01 dB |
|  | HRWP | 4.63 ± 0.04 eB | 8.47 ± 0.04 dA | 11.19 ± 0.01 aA | 9.18 ± 0.03 cA | 9.90 ± 0.01 bA |
| Yeast–Mold | NWP | 4.91 ± 0.01 eA | 8.44 ± 0.01 cB | 8.98 ± 0.03 bB | 9.20 ± 0.03 aA | 8.36 ± 0.03 dB |
|  | HRWP | 4.38 ± 0.04 eB | 9.75 ± 0.06 cA | 10.00 ± 0.06 bA | 9.36 ± 0.06 dA | 10.70 ± 0.07 aA |
| LAB | NWP | 0.00 ± 0.00 eA | 8.62 ± 0.06 cB | 9.78 ± 0.04 aB | 9.17 ± 0.08 bA | 8.03 ± 0.18 dB |
|  | HRWP | 0.00 ± 0.00 dA | 10.01 ± 0.01 bA | 11.00 ± 0.14 aA | 9.30 ± 0.42 cA | 9.00 ± 0.14 cA |

a, b, c Different lowercase letters (in the same row) indicate the presence of significant differences ($p < 0.05$) between samples in different fermentation periods. A, B, C Different uppercase letters (in the same column) indicate the presence of significant differences ($p < 0.05$) between samples in the same fermentation period.

Initially, all samples showed low microbial levels, while after one week of fermentation, a prominent increase in counts of different microbial strains was shown. All microbial strains, i.e., TMAB, yeast–molds, and LAB, reached their peak after two weeks of fermentation followed by a decline phase. All microbial strains showed counts higher in HRWPs than NWPs at all fermentation stages ($p < 0.05$). Similar results were obtained in non-controlled and naturally fermented cucumber where the highest LAB and mold–yeast

counts were obtained after about 2 and 4 weeks of fermentation, respectively [20]. The authors found that LAB counts of naturally fermented cucumber prepared by hydrogen-purged brine were higher than the non-purged one (control) after 4 days of fermentation, while yeast–mold counts were higher than those of control in the first 2 days [20].

HRWPs showed a continuous increase in yeast–mold counts compared to NWP samples, which can accelerate the spoilage of the product. The method of preparation of beet pickle used in the present study mimics that used at the home scale and is characterized by its non-controlled fermentation and non-pasteurization treatment at the end of the fermentation period. However, on an industrial scale, the fermented vegetable is generally submitted to a pasteurization process to kill the spoilage microorganisms, especially the yeasts and molds that deteriorate the product at late stages. In light of the results of the present study, we can suggest submitting the fermented product to a heat treatment at the end of the fermentation stage (1st week) followed by the addition of a new sterile acidified HRW brine.

Regarding the higher counts of microorganisms in HRWPs compared with NWs, the presence of molecular hydrogen in HRWPs might help microorganisms to better grow due to the neutralization of free radicals that were formed during growth.

### 3.3. Biogenic Amines

Aromatic amines (tyramine, 2-phenylethylamine), heterocyclic amines (histamine, tryptamine), and aliphatic di-amines (putrescine) were analyzed in both beet slices and brine of NWPs and HRWPs throughout the fermentation stages (Figure 1a,b). Levels of all studied BAs were null in brine and 0.09–0.6 mg/Kg in beet samples at the starting time, i.e., 0 days, and then increased significantly with time ($p < 0.05$). For all fermentation stages, the levels of all studied BAs were lower in HRWPs than those of NWPs ($p < 0.05$). The levels of all BAs in brines were higher than those of beets.

Initially, biogenic amines were not found in the brine samples, however, in beet samples they were detected at 0.53 and 0.56 (tyramine), 0.11 and 0.09 (2-phenylethylamine), 0.34 and 0.31 (histamine), 0.35 and 0.31 (tryptamine), and 0.56 and 0.60 (putrescine) mg/kg for NWPs and HRWPs, respectively, without significant differences between them ($p > 0.05$). Generally, all biogenic amine amounts were higher in brine than in beet samples.

### 3.3.1. Aromatic Amines
Tyramine

At the end of fermentation i.e., the 56th day, the level of tyramine was 72.76 and 61.74 mg/kg in beet and 113.49 and 92.67 mg/kg in brine for NWPs and HRWPs, respectively ($p < 0.05$). The level of tyramine decreased by 15.15 and 18.35% in beet and brine, respectively, when HRW was used in pickle preparation instead of normal water ($p < 0.05$).

Phenylethylamine

Among the BAs, 2-phenylethylamine showed the lowest levels for all samples ($p < 0.05$). At the end of fermentation, i.e., the 56th day, the level of 2-phenylethylamine was 48.00 and 40.00 mg/kg in beet and 58.01 and 50.19 mg/kg in brine for NWPs and HRWPs, respectively ($p < 0.05$). The level of 2-phenylethylamine decreased by 16.67 and 13.44% in beet and brine, respectively, when HRW was used in pickle preparation instead of normal water ($p < 0.05$).

### 3.3.2. Heterocyclic Amines
Histamine

At the end of fermentation, i.e., the 56th day, the level of histamine was 67.89 and 49.12 mg/kg in beet and 91.74 and 70.92 mg/kg in brine for NWPs and HRWPs, respectively ($p < 0.05$). The level of histamine decreased by 27.65 and 22.7% in beet and brine, respectively, when HRW was used in pickle preparation instead of normal water ($p < 0.05$).

Tryptamine

At the end of fermentation, i.e., the 56th day, the level of tryptamine was 93.14 and 77.23 mg/kg in beet and 119.00 and 93.11 mg/kg in brine for NWPs and HRWPs, respectively ($p < 0.05$). The level of tryptamine decreased by 17.09 and 21.76% in beet and brine, respectively, when HRW was used in pickle preparation instead of normal water ($p < 0.05$).

### 3.3.3. Aliphatic Di-Amines
Putrescine

At the end of fermentation, i.e., the 56th day, the level of putrescine was 81.11 and 63.56 mg/kg in beet and 106.75 and 85.93 mg/kg in brine for NWPs and HRWPs, respectively ($p < 0.05$). The level of putrescine decreased by 21.64 and 19.5% in beet and brine, respectively, when HRW was used in pickle preparation instead of normal water ($p < 0.05$).

The use of H2 water in red beet pickles (HRWPs) showed significant effects on restricting the formation of tryptamine, histamine, tyramine, putrescine, and 2-phenylethylamine, compared with control ($p < 0.05$) (Figure 1a,b).

## 4. Discussion

The formation of BAs in foods, especially fermented ones, forms a serious health hazard for consumers. Sauerkraut was investigated most frequently among all fermented vegetables for the presence of BAs. For example, the formation of BAs in spontaneously fermented Chinese sauerkraut was reported as follows: putrescine (24–45 mg/kg), cadaverine (10–35 mg/kg), and tyramine (30–38 mg/kg), whose levels increased with the fermentation time [24].

The formation of BAs was correlated with the growth of microorganisms and specifically with those producing amino acid decarboxylases. This shows that the formation of BAs in foods depends on the presence of amino acids and the proteases and peptidase activity as well as the presence and activity of decarboxylases in the medium.

It has been reported that the catabolism of amino acids by LAB can be affected by different environmental factors such as pH, salt concentration, temperature, and Eh [25]. The authors revealed that oxidative Eh conditions increased the $\alpha$-keto acid decarboxylase activity of *L. lactis* NCDO 1867. They also reported that reducing Eh significantly stimulated the production of hydroxy acids and carboxylic acids from three amino acids, i.e., phenylalanine, leucine, and methionine, by two strains of *Lactococcus lactis*, i.e., *L. lactis* NCDO 1867 and *L. lactis* NCDO 763. The authors revealed that the level of the degradation of phenylalanine (Phe), leucine (Leu), and methionine (Met) was higher under reducing conditions (similar to HRWPs) than under oxidative conditions (similar to NWPs) in *Lactococcus lactis* NCDO763. Referring to the results of the latter study, we can suggest that reducing conditions such as in HRWPs might stimulate the degradation of the substrate of decarboxylase, i.e., amino acids, that might lead to a decrease in the formation of BAs. This assumption needs to be proved by further study to evaluate the effect of Eh on the gene expression and activity of decarboxylases.

The formation of BAs was considered a protective mechanism against intracellular acidification, allowing the cell to maintain pH homeostasis through the alkalinization process during the growth in acidic conditions or as a source of metabolic energy [26]. Most bacteria utilize acid stress-induced amino acid decarboxylases to maintain pH homeostasis [27]. The formation of BAs needs to involve decarboxylase in decarboxylase-producing microorganisms, which is induced by acidic conditions [28]. These decarboxylases are induced in low-pH media in the late exponential growth phase, and the substrate, i.e., the amino acid, must be available at high concentrations (millimolar range). It has been reported that the low pH values activate histidine decarboxylase (HDC) of *Oenococcus oeni*. The synthesis of decarboxylase has been found to be maximal during the stationary phase when the pH was very low due to lactic acid accumulation [5].

Molenaar et al. (1993) studied the growth of *Lactobacillus buchneri* isolated from Swiss cheese and implicated in an outbreak of food poisoning [26]. They revealed that after the growth in a complex medium supplemented with 10 mM histidine, *L. buchneri* cells

had about fivefold higher histidine decarboxylase activity than cells grown in the absence of additional histidine. Additionally, tyrosine decarboxylase activity was increased by about twofold and tyramine production by tenfold in *L. brevis* when it had been cultivated in MRS including tyrosine [5]. Moreover, histidine has been shown to induce histamine accumulation in both *Lactobacillus* 30a and *L. hilgardii* ISE 5211. These studies show that the synthesis of decarboxylases by microorganisms in the formation process of BAs is induced by acidic conditions and the presence of amino acids.

Since LAB are generally found in an ecological system together with other microorganisms, the autolysis of microorganisms generates free proteins that undergo proteolytic breakdown by LAB which possess complex proteolytic and peptidolytic systems (membrane-bound or extracellular proteases with pH optima of 5.5–6.5, and intracellular peptidases) allowing an increase in the release of amino acids in the medium [5]. Moreover, Capuani et al. (2014) reported that reducing conditions (performed by bubbling N2/H2, 95/5) did not affect the proteolysis in all fermentations of buckwheat sourdough except for *Weissella cibaria* [29].

Decarboxylase activity is often expressed independently of cell viability and these enzymes also maintain their activity after cell lysis in harsh environmental conditions [6]. This can explain the results of the present study showing the increasing levels of BAs during all the long fermentation periods, including the late ones where a large part of LAB cells may have been submitted to the lysis process.

According to the above-cited studies, it seems that the possible explanation for the restrictive effect of HRW on the formation of BAs may be related to the impact of molecular hydrogen/lower redox values on the synthesis and/or the activity of decarboxylases (See Graphical Abstract).

It was reported that the introduction of additional disulfide bonds in the active site of decarboxylase led to an increase in its activity or its stability [27]. Indeed, according to this report, an increasing release of oxidant substances during the fermentation process allowing an increase in the Eh value of the medium (such as in NWP samples) might lead to an increase in the stability/activity of decarboxylases by oxidizing the sulfhydryl group (-SH) of the decarboxylase proteins by forming disulfide bonds [27]. The sulfhydryl group of amino acids is a nucleophile and reductant that can scavenge free radicals, leading to transformation into a disulfide group [30,31]. The formation of oxidizing substances such as free radicals during microbial growth might lead to the formation of disulfide bonds in decarboxylase protein, allowing it to increase its activity and stability. Molecular hydrogen (such as in HRWP samples) with its reducing and antiradical properties in the medium might scavenge some of these oxidizing substances such as the hydroxyl radical (●OH) and peroxynitrite (ONOO-) [32]. Additionally, Wu et al. (2017) revealed that molecular hydrogen more effectively prevented $H_2O_2$-induced IP3R1 dysfunction by reducing disulfide bonds rather than quenching ROS [30]. $H_2$ potentially altered the conformation of the IP3R1 by breaking disulfide bonds. The authors correlated the protective effect of $H_2$ with the reduction of disulfide bond formation in the protein caused by oxidative stress, and not by quenching ROS. Additionally, the authors mentioned that $H_2$, even at a lower concentration, can disrupt metabolic oxidoreduction reactions. Furthermore, $H_2$ has also been reported to induce superoxide dismutase (SOD) and heat shock protein (HSP) activity to quench ROS production [30]. The authors suggested that the activation of glutathione/thioredoxin systems underlying the H2-induced elimination of ROS damage of IP3Rs allows a decrease in the $H_2O_2$-induced disulfide bond formation. Similarly, Keten et al. (2012) reported an in silico study in which the stability of the disulfide bond was strongly influenced by the redox potential of the medium [33]. The presence of $H_2$ led to breaking the protein disulfide bond by reducing the force required, while $O_2$ did not show any rupture effect. However, the authors reported that the rupture of disulfide bonds was dependent on the duration of exposition to the reducing agent ($H_2$), where the weakening effect of the reducing environment was only seen at longer durations. The

previous statement correlates with the conditions of HRWP samples where $H_2$ found in HRW might stay for a long time inside the jars.

Lin et al. (2022) performed a bioinformatic analysis of the amino acid sequences of four amino acid decarboxylases (AADs) of common microorganisms found in pickle fermentation [34]. The authors found that most AADs were cytosolic endoenzymes. They revealed that different spices such as ginger and pepper showed higher component content with a strong binding ability to AADs. While the authors correlated this inhibition to the higher number of hydrogen bonds and hydrophobic interactions between the spice components and the active site of AADs than between the precursor AA and AADs, they assumed that AADs may also be inhibited by the change in the oxidoreduction potential (low Eh) of the medium due to reducing compounds such as phenolics from the spices. The authors assumed that the high cysteine content found in AADs (up to 15) may form -SH or S–S groups, which are susceptible to the change in the Eh value of the medium [35]. This study relays the Eh of the medium with the protein structure of the decarboxylases, especially the -SH/S–S balance.

It was reported that anaerobic conditions favor the decarboxylase activity of *Lactobacillus curvatus* strain CTC273 [36]. In another study, the growth of *Lactococcus lactis* subsp. *lactis* and *L. lactis* subsp. *cremoris* led to higher levels of tyramine under anaerobic conditions [7]. This shows that the anaerobic conditions promote the formation of BAs by LAB. It is interesting that, in the present study, although HRWP samples had anaerobic conditions similar to NWPs, a restriction of biogenic amine formation could be obtained only in HRWP samples.

Our results showed that the levels of putrescine and tryptamine increased progressively with time in both red beet pickles and brine ($p < 0.05$), and their levels were higher in control samples than those containing HRW ($p < 0.05$). In parallel to our study, the presence of molecular hydrogen in washing water significantly decreased the formation of BAs in LAB-cultured butter [18].

The results of the present study show that the conventional commercial method used for the preparation of pickles exhibits a risk of BA formation during the fermentation phase even when producers change the fermentation medium for a new brine because, according to our results, the product itself contains BAs formed during the fermentation stage. However, the at-home method has a higher risk of BA formation due to the long period of the fermentation stage with active LAB due to the absence of heat treatment of the product. Thus, the use of HRW at the fermentation stage can limit BA formation. The formed pickles should be pasteurized in a new acidified and brined HRW.

## 5. Conclusions

Biogenic amines are frequently encountered in fermented products such as pickles. The control of the formation of biogenic amines forms a serious challenge to both processors and consumers. Many methods were proposed to keep the levels of BAs in foods under risk limits. In the last decade, many studies showed the health benefits of HRW. Our team could reveal many benefits of HRW when it was used in food preparation. In the present study, it was observed that the use of HRW in the preparation of red beet pickles could significantly restrict the formation of BAs. All the levels of BAs were lower in pickles prepared with hydrogen-rich water than those prepared with the conventional method, i.e., drinking water. Since the saturation limit of dissolved hydrogen in water is low, i.e., 1.6 mg/L in normal room conditions, a small volume of hydrogen gas is needed for preparing HRW. The results of this study suggest that the best method for limiting the formation of biogenic amines in pickles is to use HRW as a fermentation and filling medium. At the end of the fermentation stage, the fermented vegetables should be immersed in a new acidified and brined HRW followed by a pasteurization process. Additionally, no extra equipment and no energy are needed for preparing HRW. This shows that the tested method is inexpensive and does not leave toxic residues in the product and the environment, which can mark this method as a green technology.

**Author Contributions:** Conceptualization, M.B. and Y.Ç.; Methodology, M.B. and Y.Ç.; Software, Y.Ç.; Validation, Y.Ç.; Formal Analysis, Y.Ç.; Investigation, D.A.; Writing—Original Draft Preparation, Y.Ç. and D.A.; Writing—Review and Editing, D.A.; Supervision, D.A.; Project Administration, D.A. All authors have read and agreed to the published version of the manuscript.

**Funding:** This research received no external funding.

**Institutional Review Board Statement:** Not applicable.

**Informed Consent Statement:** Not applicable.

**Data Availability Statement:** The data presented in this study are available on request from the corresponding author.

**Conflicts of Interest:** The authors declare no conflict of interest.

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
