# Peer review of "Hydrogen-Rich Water Can Restrict the Formation of Biogenic Amines in Red Beet Pickles"

_fermentation, doi:10.3390/fermentation8120741_

Round 1

Reviewer 1 Report

The first question for the Authors concerns the existence of safe limit of BAs concentration in fermented food products, in particular fermented plant based products? If these limits exist, please consider the possibility to implement this information in the Introduction or in 3.3 paragraph.

The graphs in Figure 2 (paragraph 3.3) have to be re-edited if possibile.

Line 315 change hemostasis with homeostasis

The Authors are requested to emphasize the importance of applying hydrogen enriched water in an industrial process.

Author Response

Question: The first question for the Authors concerns the existence of safe limit of BAs concentration in fermented food products, in particular fermented plant based products? If these limits exist, please consider the possibility to implement this information in the Introduction or in 3.3 paragraph.

Answer: Thank you for this insightful comment. A new paragraph related to this issue was added in the introduction section.

Question: The graphs in Figure 2 (paragraph 3.3) have to be re-edited if possibile.

Answer: The graphs were re-edited.

Question: Line 315 change hemostasis with homeostasis

Answer: accepted and corrected

Question: The Authors are requested to emphasize the importance of applying hydrogen enriched water in an industrial process.

Answer: Thank you for this insightful comment. A new paragraph related to this issue was added in the introduction section.

Reviewer 2 Report

The topic of this research is really interesting and quite new, the research team investigating the effect of adding hydrogen on the fermentation and biogenic amines production during fermentation. The introduction is well done, with enough literature and e very good description of the research aim. Results and discussion are enough.

Some proposals for the authors to improve their manuscript:

-          In Materials and methods, at point 2.3. Microbiological counts, please explain better the content of the lines 92-97. It is not at all clear why you used the Stomacher machine and how you made the inoculation;

-          Why you used both, table 1 and figure 1, to offer the same results on the pH and Eh7? Only one of them is enough, please justify your decision if you maintain both ways of describing the results;

-          The initial number of LAB is 0, how you explain their apparition during the fermentation?

-          Some references are not clear, as “According to the previously cited studies…” (line 349) or “according to the latter report” (line 355) who is actually the same report, meaning the reference [26]. Such phrases have to be reformulated;

-          The expression “On the other hand” is extensively used here, please reformulate.

Author Response

Some proposals for the authors to improve their manuscript:

Question:      In Materials and methods, at point 2.3. Microbiological counts, please explain better the content of the lines 92-97. It is not at all clear why you used the Stomacher machine and how you made the inoculation;

Answer: This section was re-edited with additional information.

Question:          Why you used both, table 1 and figure 1, to offer the same results on the pH and Eh7? Only one of them is enough, please justify your decision if you maintain both ways of describing the results;

Answer: Thank you for your insightful comment. We deleted Figure 1 from the manuscript. 

Question:          The initial number of LAB is 0, how you explain their apparition during the fermentation?

It was reported that the percentage of LAB counts in fresh vegetables is too low compared to the other spoilage microorganisms, and its percentage ranges between 0.01 to 0.01%. However, during the fermentation stage, due to the presence of NaCl in the brine at high levels (2-8% according to the product), only LAB can grow and its counts increase while spoilage microorganisms are inhibited.

Question:      Some references are not clear, as “According to the previously cited studies…” (line 349) or “according to the latter report” (line 355) who is actually the same report, meaning the reference [26]. Such phrases have to be reformulated;

Answer: We changed the statement "According to the previously cited studies" to "According to the above-cited studies". In fact, this statement means that the above-discussed paragraphs with their references conclude the statement provided after the sentence "According to the previously cited studies".

Regarding the statement "according to the latter report” (line 355) ", as there is only one reference in the same paragraph that precedes this sentence, this means that the concerned reference is number 26 because before this sentence there in only this reference in the same paragraph. 

Question:          The expression “On the other hand” is extensively used here, please reformulate.

Answer: accepted and corrected